# Construction of an Integrated Drought Monitoring Model Based on Deep Learning Algorithms

Yonghong Zhang [1,*], Donglin Xie [1], Wei Tian [2], Huajun Zhao [1], Sutong Geng [1], Huanyu Lu [1], Guangyi Ma [3], Jie Huang [1] and Kenny Thiam Choy Lim Kam Sian [4]

1   School of Automation, Nanjing University of Information Science and Technology, Nanjing 210044, China
2   School of Computer Science, Nanjing University of Information Science and Technology, Nanjing 210044, China
3   School of Electronics and Information Engineering, Nanjing University of Information Science and Technology, Nanjing 210044, China
4   School of Atmospheric Science and Remote Sensing, Wuxi University, Wuxi 214100, China
*   Correspondence: zyh@nuist.edu.cn

**Abstract:** Drought is one of the major global natural disasters, and appropriate monitoring systems are essential to reveal drought trends. In this regard, deep learning is a very promising approach for characterizing the non-linear nature of drought factors. We used multi-source remote sensing data such as the Moderate Resolution Imaging Spectroradiometer (MODIS) and Climate Hazards Group Infrared Precipitation with Station (CHIRPS) data to integrate drought impact factors such as precipitation, vegetation, temperature, and soil moisture. The application of convolutional long short-term memory (ConvLSTM) to construct an integrated drought monitoring model was proposed and tested, using the Xinjiang Uygur Autonomous Region as an example. To better compare the monitoring performance of ConvLSTM models, three other classical deep learning models and three classical machine learning models were also used for comparison. The results show that the composite drought index (CDI) output by the ConvLSTM model had a consistent high correlation with the drought rating of the multi-scale standardized precipitation evapotranspiration index (SPEI). The correlation coefficients between the CDI and the multi-scale standardized precipitation index (SPI) were all above 0.5 ($p < 0.01$), which was highly significant, and the correlation coefficient between CDI-1 and the monthly soil relative humidity at a 10 cm depth was above 0.45 ($p < 0.01$), which was well correlated. In addition, the spatial distribution of the CDI-6 simulated by the model was highly correlated with the degree of drought expressed by the SPEI-6 observations at the stations. This study provides a new approach for integrated regional drought monitoring.

**Keywords:** drought monitoring; Xinjiang; MODIS; ConvLSTM

## 1. Introduction

Drought is a common natural disaster that covers a wide area, occurs frequently, and lasts for a long time. Thus, it poses a serious threat to crop production, the ecological environment, and sustainable socioeconomic development [1,2]. Droughts are classified as meteorological, agricultural, hydrological, and socioeconomic droughts [3]. At present, drought monitoring methods include meteorological monitoring methods based on station data and remote sensing methods based on remote sensing data [4]. Although station-data-based meteorological monitoring methods are relatively mature, they cannot be used to effectively monitor large areas due to the uneven distribution and limited quantity of station data. They have certain limitations when applied to regional-scale drought monitoring [5]. Compared with station data, remote sensing data have the advantages of high resolutions, long time series, and wide ranges [6]. Therefore, using remote sensing data to construct drought monitoring models has become an important research direction.

Many drought monitoring indicators have been proposed, mainly including traditional indicators based on station meteorological data and remote sensing indicators based on satellite data, which provide quantitative information on drought severity [7]. For example, the main traditional indicators based on station meteorological data are the standardized precipitation index (SPI) [8], the Palmer drought severity index (PDSI) [9], and the standardized precipitation evapotranspiration index (SPEI) [10]. Among these, SPEI is widely used in the meteorological community to monitor drought severity. SPEI is relative to PDSI and SPI in that it is calculated from precipitation and temperature and has more comprehensive information coverage than PDSI and SPI. SPEI has the advantages of PDSI and SPI while incorporating multiple time scales. As a result, SPEI is an effective indicator for monitoring and assessing global drought changes. However, the SPEI calculated based on station data is not spatially representative and has some limitations when applied to regional-scale drought monitoring [11].

In recent years, scholars have proposed the use of remote sensing drought indices to study drought conditions at larger spatial scales, including the normalized difference vegetation index (NDVI) [12], the temperature condition index (TCI) [13], and the precipitation condition index (PCI) [14]. As drought is affected by many factors and traditional remote sensing monitoring mainly monitors single factors such as vegetation, precipitation, and temperature, it cannot fully reflect drought information [15]. Therefore, many scholars have proposed using fused multi-source remote sensing indices to monitor drought in order to solve these problems. For example, Aisyah et al. [16] used a polynomial equation to construct a feature space from the normalized difference vegetation index (NDVI) and the land surface temperature (LST) to establish a temperature vegetation drought index (TVDI) to monitor regional drought. Yuan et al. [17] used apparent thermal inertia (ATI) and TVDI to estimate soil moisture on the Loess Plateau of China. Kubiak et al. [18] evaluated the relationship between precipitation and hydrological conditions and the relationship between the two drought types using SPI as a meteorological drought indicator and the standardized water level index (SWI) and standardized runoff index (SRI) as hydrological drought indicators.

Current studies mainly use classical regression methods to construct drought monitoring models. For example, Chen et al. [19] used a linear regression to construct a meteorological composite drought index for the spatial and temporal assessment of drought in Hubei Province. Xun et al. [20] used a multi-source linear regression combined with potential evapotranspiration (PET) and soil moisture to construct a model for drought monitoring in Henan Province. However, in cases with more types of remotely sensed drought indicators, linear regression methods can lead to difficult modeling when the relationships between variables become complex.

With machine learning (ML) becoming popular, some scholars attempted to use data mining methods to build drought monitoring models. Zhang et al. [21] used the gradient boosting machine (GBM) and the extreme randomized tree (ERT) algorithm to model the overall drought conditions in China. Kau et al. [22] used classical machine learning models such as artificial neural networks (ANN), support vector machines (SVM), and random forests (RF) to assess and predict drought. Hamade et al. [23] used a random forest model that considered the vegetation condition index (VCI), the temperature condition index (TCI), and other remotely sensed drought indices to construct an integrated drought monitoring model. The results showed that machine learning has greatly improved data mining and prediction accuracy compared to traditional linear regressions. However, with the increases in remote sensing data and drought impact factors, there are limitations in the ability of machine learning to extract information from various factors [24], and more research is needed to address this aspect.

Deep learning is a neural-network-based machine learning method that was proposed by Hinton et al. [25] and can simulate the human brain to manipulate data and outperform other machine learning models. It has been successfully applied to solve problems such as computer vision, natural language processing, speech recognition, and energy

prediction [26]. In addition, deep learning algorithms have the ability to extract more useful features from a large number of drought factors, which has a significant effect on the construction of comprehensive drought monitoring models [27]. However, there are few studies using deep learning algorithms for drought monitoring. Shen et al. [28] used deep forward neural networks (DFNN) combined with multi-source remote sensing data to assess agricultural drought. Although they used a simple deep learning method to construct a drought monitoring model, there were still problems, such as a single type of model and poor feature extraction. To address these problems, we considered the respective characteristics of convolutional neural networks (CNNs) and LSTM networks. CNNs have the advantage of strong feature extraction capabilities [29], while LSTM excels in mining time series data [30]. In order to make full use of their advantages, we combined these two models to obtain a new and effective model.

Therefore, we proposed an integrated drought monitoring model based on the ConvL-STM network. It also used meteorological precipitation, soil moisture, surface temperature, and vegetation growth as independent variables and station SPEI values as dependent variables for drought monitoring, which had not been previously studied in Xinjiang. To compare model performance, we also used six models for comparison, including long short-term memory (LSTM), a convolutional neural network (CNN), a deep forwarded network (DFNN), random forest (RF), a support vector machine (SVM), and XGBoost. The main objectives of this study were (1) to construct a drought monitoring model using ConvLSTM and to compare the monitoring performance with other benchmark models; (2) to validate the ability of the model to output a comprehensive drought index (CDI) to monitor other drought indices and analyze the relative importance of each drought factor to the CDI; and (3) to produce a simulation of the spatial and temporal changes in regional drought trends in Xinjiang during a typical drought year (March–August 2014) using CDI.

## 2. Materials and Methods

### 2.1. Study Area

Xinjiang, one of the driest regions in the world, is located in the northwestern region of China, with a geographical range between 73°29′54″–96°23′3″ E and 34°20′11″–49°10′55″ N. It has a land area of 164,011.03 km$^2$ and is the largest province in China in terms of land area [31]. The province has a variety of landforms, with the Altai Mountains in the north, the Kunlun Mountains-Alishan in the south, the Tianshan Mountains running through the central part of Xinjiang, and the Junger Basin and Tarim Basin between the three mountains (Figure 1). Due to its geographical location and topography, Xinjiang has a typical temperate continental climate, with short summers and long cold winters, and large differences between the minimum and maximum temperatures. Due to the scarcity of water resources and low rainfall, drought has become a major natural disaster in Xinjiang.

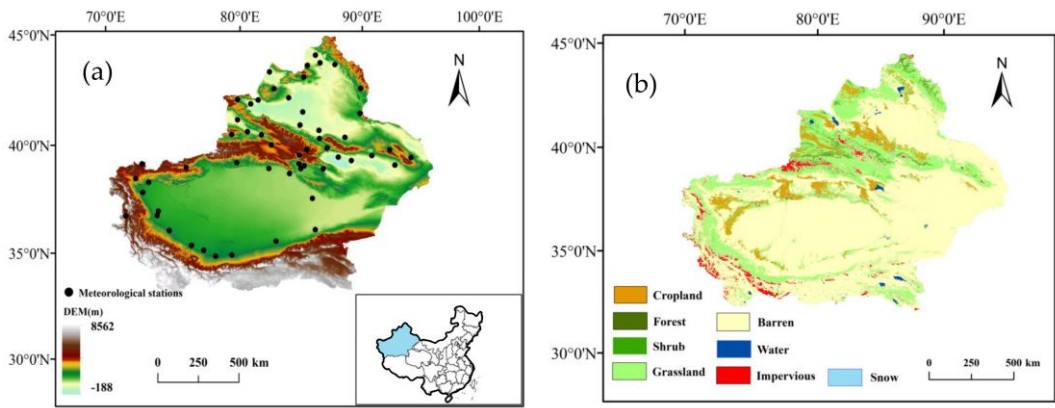

**Figure 1.** Map of the study area. (**a**) Elevation map and locations of meteorological stations. (**b**) Land cover map in 2010.

### 2.2. Data

In this study, drought indices were calculated using remote sensing data as input parameters for a deep learning model. A detailed description of the datasets is shown in Table 1.

**Table 1.** Details of the remote sensing datasets used in this study.

| Data Sources | Data Type | Variables | Temporal Resolution | Spatial Resolution | Coverage |
|---|---|---|---|---|---|
| MODIS | MOD13A1 | NDVI | 16 days | 500 m | Global |
| | MOD16A2 | ET | 8 days | 500 m | Global |
| | MOD11A1 | LST | daily | 1000 m | Global |
| | MOD15A2H | LAI | 8 days | 500 m | Global |
| UCSB-CHG | CHIRPS | Precipitation | Monthly | $0.25° \times 0.25°$ | Global |
| GLDAS | GLDAS-2.1 | Soil moisture | Monthly | $0.25° \times 0.25°$ | Global |

#### 2.2.1. MODIS Data

MODIS (Moderate Resolution Imaging Spectroradiometer) is a multispectral medium/high-resolution sensor that is carried on the Terra and Aqua satellites and provides a wide range of biophysical and environmental products and is widely utilized [32]. In this study, we used MOD13A1 as the vegetation index product. MOD13A1 is a 16-day synthetic normalized vegetation index (NDVI) with a spatial resolution of 500 m [33]. The MOD11A1 product is a daily synthetic surface temperature (LST) with a spatial resolution of 1 km. These two products were used to calculate the vegetation condition index (VCI) and the temperature condition index (TCI). In addition, the MOD16A2 product has an 8-day synthetic spatial resolution of 500 m for evapotranspiration (ET) [34], and the MOD15A2H product has an 8-day synthetic spatial resolution of 500 m for the leaf area index (LAI) [35]. The MODIS data were from https://ladsweb.modaps.eosdis.nasa.gov/ (accessed on 27 May 2022). The data were for the period of 2000–2020 and were averaged to produce monthly data.

#### 2.2.2. CHIRPS Data

CHIRPS is a dataset with records of global rainfall from 1981 to the present. CHIRPS combines satellite imagery at a $0.05°$ resolution with in situ data to create gridded rainfall time series for precipitation trend analysis and seasonal drought monitoring. This product was used to calculate the precipitation conditions index (PCI) [36]. The CHIRPS data for the period of 2000–2020 were downloaded from https://www.chc.ucsb.edu/data/chirps (accessed on 27 May 2022).

#### 2.2.3. GLDAS Data

The GLDAS data are a data assimilation product that was jointly developed by NASA's Goddard Space Flight Center (GSFC) and NOAA's National Centers for Environmental Prediction (NCEP). They combine soil moisture data from four land surface models (Mosaic, Noah, CLM, and VIC). The GLDAS data were used to calculate the soil moisture condition index (SMCI) [37]. The data were obtained from https://ldas.gsfc.nasa.gov/gldas (accessed on 27 May 2022) for the period of 2000–2020.

#### 2.2.4. Meteorological Station Data

The raw meteorological observation station data were obtained from the National Meteorological Information Centre (http://data.cma.cn/) (accessed on 27 May 2022), and the daily temperature and precipitation data from 2000-2020 were selected from the national benchmark stations in Xinjiang. The daily temperature data were averaged, and the precipitation data were summed to obtain the monthly data. The records of stations with

more than three consecutive months of missing measurements were excluded, and the observation records of 55 meteorological stations were finally obtained [38]. The monthly SPEI time series were calculated for each station for the 1-,3-, 6-, and 12-month time scales from 2000 to 2020.

*2.3. Baseline Model*

2.3.1. Machine Learning Models

Random forest (RF) is an integrated method based on classification and regression trees (CART) that overcomes the major limitations of CART by aggregating multiple independent trees. RF is a relatively new machine learning algorithm with a fast learning process, fast computing speed, good stability, efficiency in processing large datasets, and high prediction accuracy that is less prone to overfitting [39]. Therefore, we used RFs as a classical machine learning method to compare and validate deep learning models. XGBoost offers higher accuracy through the introduction of second-order Taylor expansions. At the same time, XGBoost's base learner can be either a decision tree or a linear classifier, for greater flexibility and support column sampling, which reduces overfitting and computation [40]. Support vector machine (SVM) is a supervised learning method that uses the principle of structural risk minimization to map low-dimensional space and linearly indistinguishable data to high-dimensional space to make them linearly distinguishable through non-linear mapping and then classifies and predicts the data in the high-dimensional space [41]. Therefore, we used these three classical machine learning and deep learning models for comparison and validation.

2.3.2. Deep Forwarded Neural Network (DFNN)

The deep forwarded neural network (DFNN) is a classical deep learning research model that can extract relevant features from a large number of input variables to obtain high-accuracy predictions in regression tasks [42]. The model structure is divided into three main parts: the input layer, the hidden layer, and the output layer. In addition, in order to avoid overfitting during training, a dropout layer was added between the input and hidden layers to achieve a better generalization performance by randomly discarding some neurons [43]. During the DFNN model's training, model optimization was achieved by adjusting the hidden layers, the neurons, and the number of iterations. Cross-validation was used to verify the model's performance between the test and training sets.

2.3.3. Convolutional Neural Network (CNN)

The convolutional neural network (CNN) is a class of neural networks that include convolutional computation and have a deep structure with powerful automatic feature extraction capabilities. They are representative algorithms for deep learning [44]. The network usually consists of multiple convolutional layers, pooling layers, and fully connected layers stacked on top of each other. The convolutional layers enable the extraction of data features, the pooling layers reduce the number of parameters, and the multiple extracted features are combined through a fully connected layer. The main advantage of a CNN is the use of input information features for learning, which is appropriate in cases where there are dependencies between the input data, reducing computational costs and solving the overfitting problem.

2.3.4. Long Short-Term Memory (LSTM)

Long short-term memory (LSTM), a variant structure of the recurrent neural network (RNN), is a network with a long-term memory function. Due to the problem of gradient disappearance, RNNs cannot handle time series with excessive time delays. Hu et al. [45] proposed a new structure in which the middle layer of a traditional RNN is replaced by LSTM blocks. In addition, LSTM networks are widely used for time series samples due to their characteristics and have very important roles in natural language processing, speech recognition, rainfall prediction, etc. The LSTM module consists of three parts: a

forget gate, an input gate, and an output gate. In particular, the forget gate determines which new information remains in the cell state and updates the cell state. The input gate determines what information is discarded from the cell state, and the output gate controls the output of the cell state. The important parameters with their choice of values are present in Appendix A.

### 2.4. Data Processing

Unlike other natural hazards, drought is a relatively complex event caused by various influencing factors and has trends spanning long time scales. Drought characteristics and drought-causing factors vary from region to region [46]. Therefore, this study considered the precipitation condition index (PCI), the vegetation condition index (VCI), the temperature condition index (TCI), the vegetation water supply status index (VSWI), the leaf area index (LAI), the vegetation health index (VHI), evapotranspiration (ET), and the soil moisture condition index (SMCI) as independent variable data for the deep learning model. The multi-scale SPEI calculated from meteorological stations was also used as the dependent variable and was input into the model together with the independent variable data to construct the comprehensive drought index (CDI). The formulae and descriptions for each input parameter variable are given in Table 2.

**Table 2.** Descriptions of the input variables.

| Type of Variable | Factors | Drought Index | Formula | References |
|---|---|---|---|---|
| Independent variables | Precipitation | PCI | $PCI = \frac{P_i - P_{min}}{P_{max} - P_{min}}$ (where $P_i$ is the monthly precipitation and $P_{max}$ and $P_{min}$ are the monthly maximum and minimum precipitation) | [47] |
| | Vegetation | VCI | $VCI = \frac{NDVI_i - NDVI_{min}}{NDVI_{max} - NDVI_{min}}$ (where $NDVI_i$ is the monthly NDVI value and $NDVI_{min}$ and $NDVI_{max}$ are the monthly minimum and maximum NDVI values) | [48] |
| | | VHI | $VHI = \alpha VCI + (1 - \alpha) TCI$ ($\alpha$ denotes a constant value set to 0.5) | [49] |
| | | VSWI | $VSWI = \frac{NDVI}{LST}$ | [50] |
| | Temperature | TCI | $TCI = \frac{LST_i - LST_{min}}{LST_{max} - LST_{min}}$ (where $LST_i$ is the monthly LST value and $LST_{max}$ and $LST_{min}$ are the monthly maximum and minimum values) | [51] |
| | Soil | SMCI | $SMCI = \frac{SM_i - SM_{min}}{SM_{max} - SM_{min}}$ (where $SM_i$ is the monthly SM value $SM_{min}$ and $SM_{max}$ are the monthly minimum and maximum SM values) | [52] |
| Dependent variables | | SPEI-1 SPEI-3 SPEI-6 SPEI-12 | $w - \frac{c_0 + c_1 w + c_2 w^2}{1 + d_1 w + d_2 w^2 + d_3 w^3}$ (w is defined as climatic water balance calculated based on the difference between precipitation and reference evapotranspiration, and $c_0$, $c_1$, $c_2$, $d_1$, $d_2$, and $d_3$ are constants.) | [53] |

TCI indicates the stressfulness of temperature on vegetation growth, with higher TCI values indicating more severe drought. VCI indirectly reflects the severity of drought by monitoring the increase in vegetation, with higher values of VCI indicating more vegetation

and less drought. In addition, VSWI is calculated from NDVI and LST. When vegetation is affected by drought, it closes some of its stomata to conserve leaf water content, resulting in lower evaporation of moisture from the vegetation. Therefore, according to the calculation principle of VSWI, it is only related to NDVI and LST in the current period, and has no relationship with historical data, which can directly reflect the current degree of drought. The smaller the value of VSWI, the smaller the drought intensity. VHI is calculated from VCI and TCI and has the characteristics of both indices. PCI is calculated from CHIRPS data and can directly respond to precipitation anomalies. SMCI directly reflects soil moisture and can quantitatively portray the degree of dryness and wetness anomalies in the soil. Each factor reflects drought information differently, so they were used as input variables for the integrated drought monitoring model.

SPEI is calculated on 1-, 3-, 6-, and 12-month time scales using the degree of difference between the precipitation and evapotranspiration to determine deviations from the average state to characterize the degree of drought in a given region. In addition, SPEI reflects different drought conditions, with SPEI-1 and SPEI-3 reflecting meteorological droughts on a scale of one to three months, SPEI-6 reflecting agricultural droughts on a scale of six months, and SPEI-12 reflecting hydrological droughts on a scale of 12 months [53]. The SPEI drought classification is shown in Table 3.

**Table 3.** SPEI's classification criteria for grading drought.

| Drought Grade | Drought Condition | SPEI |
|:---:|:---:|:---:|
| I | No drought | $-0.5 < \text{SPEI}$ |
| II | Light drought | $-1.0 < \text{SPEI} \leq -0.5$ |
| III | Moderate drought | $-1.5 < \text{SPEI} \leq -1.0$ |
| IV | Severe drought | $-2.0 < \text{SPEI} \leq -1.5$ |
| V | Extreme drought | $\text{SPEI} \leq -2.0$ |

*2.5. Convolutional Long Short-Term Memory (ConvLSTM)*

The combination of CNN and LSTM provides an excellent solution to the time series prediction problem [54]. The structure of ConvLSTM (Figure 2) is similar to that of LSTM, which consists of a storage unit and three gates (i.e., a forget gate, an input gate, and an output gate). The main difference between LSTM and ConvLSTM is that the internal matrix multiplication of LSTM is replaced by convolutional operations. On one hand, the CNN acts as the upper layer of the ConvLSTM model. It can extract complex features from the model's input variables, apply convolution operations to the incoming data, and pass the results to subsequent layers. On the other hand, convolutional operations enhance feature extraction and reduce the number of parameters. The LSTM layer is the lower layer of the ConvLSTM and supports time series prediction. As a result, the ConvLSTM model proposed in this study can extract complex drought features from multi-source remote sensing data, store complex irregular trends, and create a comprehensive drought index (CDI) by combining station SPEI with multi-source remote sensing indices to provide accurate monitoring results for the study area. The main equations are as follows:

$$i_t = \sigma(W_{xi} * X_t + W_{hi} * H_{t-1} + W_{ci} \circ C_{t-1} + b_i) \tag{1}$$

$$f_t = \sigma\left(W_{xf} * X_t + W_{hf} * H_{t-1} + W_{cf} \circ C_{t-1} + b_f\right) \tag{2}$$

$$C_t = f_t \circ C_{t-1} + i_t \circ tanh(W_{xc} * X_t + W_{hc} * H_{t-1} + b_c) \tag{3}$$

$$O_t = \sigma(W_{xo} * X_t + W_{ho} * H_{t-1} + W_{co} \circ C_t + b_o) \tag{4}$$

$$H_t = O_t \circ tanh(C_t) \tag{5}$$

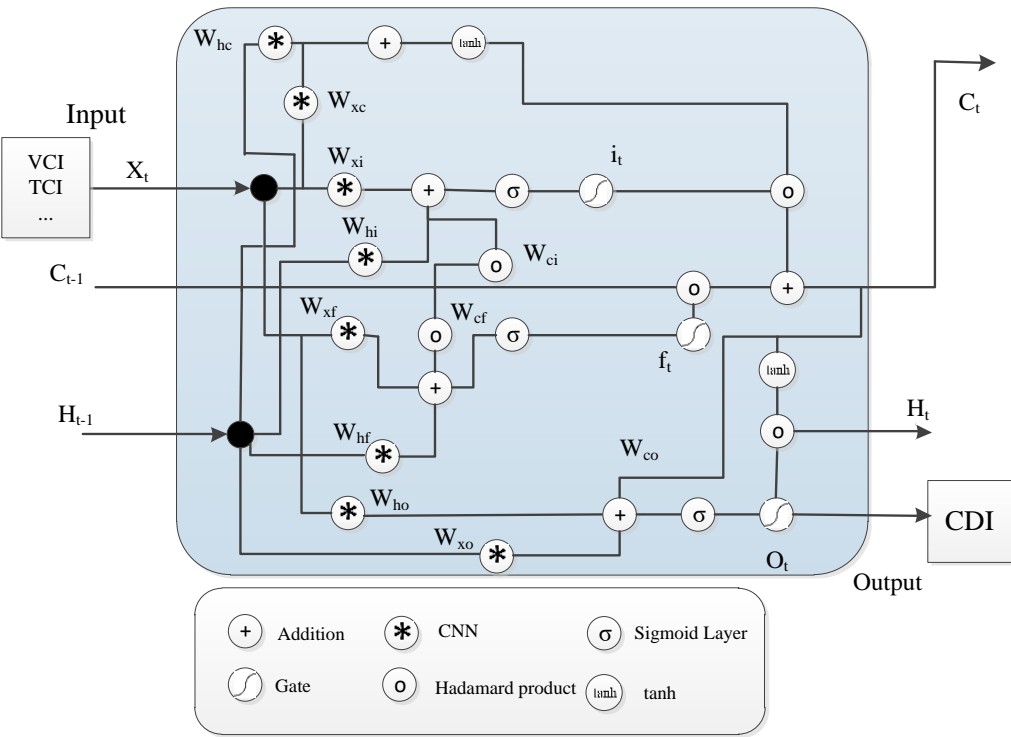

**Figure 2.** ConvLSTM internal network architecture. X, H, C, i, f, and o are the input sequence, hidden state, memory cell, input gate, forget gate, and output gate, respectively.

From the internal structure of the ConvLSTM network shown in Figure 2, $i_t$ and $O_t$ are the input and output gates, ft is the forget gate, $C_t$ is the memory cell, W represents the weight between connected neurons, $C_t$ is the cumulative state information, and σ is the sigmoid non-linear activation function that maps the output to a 0 to 1 distribution for easy convergence. A more detailed explanation of ConvLSTM can be found in [55].

### 2.6. The Process of Building the Model

The construction of an integrated drought monitoring model requires the consideration of multiple influencing factors, as drought is not only related to soil moisture and vegetation growth conditions but also to precipitation and surface temperature conditions. Therefore, in this study, eight drought-influencing factors, namely VCI, TCI, PCI, SMCI, VSWI, LAI, ET, and VHI, were used as independent variables, and SPEI at different scales was used as the dependent variable to construct an integrated drought monitoring model based on multi-source remote sensing data. In addition, we divided the constructed datasets into training (2000–2016) and testing (2017–2020) datasets. The trained models were used in the test set, and the correlation coefficients ($R^2$), root-mean-square errors (RMSEs), and mean absolute errors (MAEs) of the seven models were calculated to assess the performances of the models. The relative importance of each drought influence factor on the CDI was then analyzed. Finally, the spatial distribution of CDI-6 was plotted based on the drought events in a typical drought year to spatially validate the integrated drought monitoring model. A detailed description of the model construction process is shown in Figure 3.

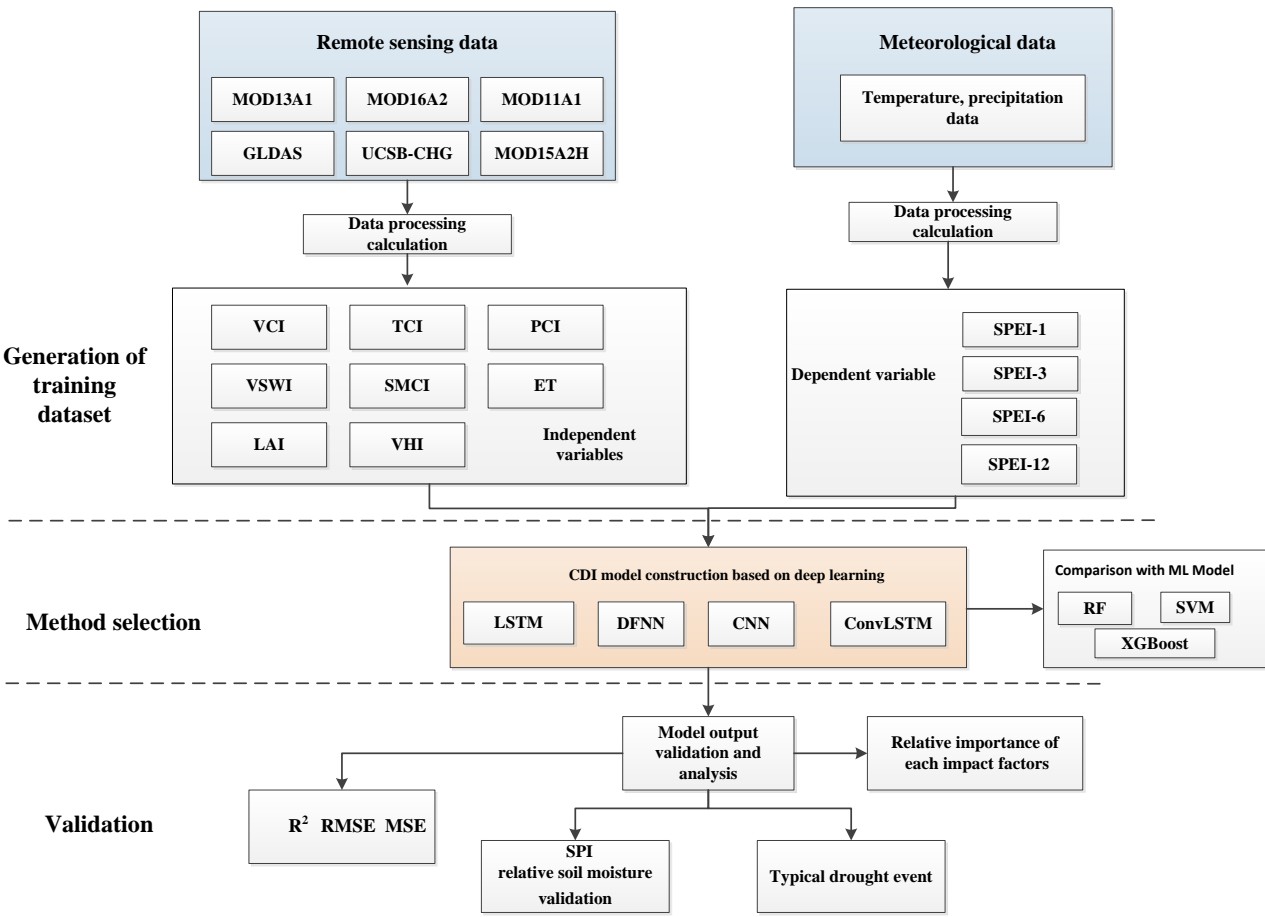

**Figure 3.** Flowchart of the drought monitoring model construction.

*2.7. Assessment Indicators*

To evaluate the performance of each prediction model, we used the correlation coefficient ($R^2$), the root-mean-square error (RMSE), and the mean absolute error (MAE).

$$R^2 = \left( \frac{\sum_{i=1}^{m}(x_i - \bar{x})(y_i - \bar{y})}{\sqrt{\sum_{i=1}^{m}(x_i - \bar{x})^2}\sqrt{\sum_{i=1}^{n}(y_i - \bar{y})^2}} \right)^2 \tag{6}$$

$$RMSE = \sqrt{\frac{\sum_{i=1}^{m}(x_i - y_i)^2}{m}} \tag{7}$$

$$MAE = \frac{\sum_{i=1}^{m}|x_i - y_i|}{m} \tag{8}$$

$R^2$ is generally used to assess the degree of conformity between the predicted and actual values, RMSE is used to measure the deviation between the predicted and actual values of the model deviation, and MAE can reflect the actual situation of the predicted value error [56]. $x_i$ denotes the CDI value of the model output, $y_i$ denotes the SPEI value, $\bar{x}$ denotes the mean of the CDI, $\bar{y}$ denotes the mean of the SPEI, and m denotes the sample size. $R^2$ values closer to 1 and RMSE and MAE values closer to 0 indicate better model performance.

*2.8. Correlation of a Single Remote Sensing Drought Index with Station SPEI*

As drought factors such as precipitation, soil, and vegetation behave differently at different time scales, remote sensing index values were extracted based on the locations of the ground-based meteorological stations during the selected time period in order to assess

the ability of a single remote sensing index to monitor drought. In this study, the correlation coefficients ($R^2$) between each single remote sensing drought index and the station drought index SPEI were calculated to analyze the ability of a single remote sensing drought index to monitor drought and the need to integrate data from multiple sources. The results are shown in Table 4. All remote sensing drought indices showed positive correlations with SPEI in general, with PCI having the highest correlation with SPEI-1, indicating that PCI is the most sensitive in monitoring short-term drought, and $R^2$ decreased with increasing SPEI time scales. SMCI had the highest $R^2$ value with SPEI-1 compared to other remote sensing indicators, suggesting that SMCI provides reliable information for monitoring short-term meteorological and agricultural drought. Similar to PCI, the correlation between TCI and SPEI-1 was higher than that of SPEI at other scales. In addition, the $R^2$ value between VCI and SPEI-12 was higher than that between VCI and SPEI at other scales, suggesting that drought information reflected in VCI has a longer lag time than that reflected in SMCI. LAI, ET, VHI, and VSWI also have the same pattern as VCI.

**Table 4.** Correlation coefficient ($R^2$) values of individual remotely sensed drought indices with different time scales of SPEI.

|  | VCI | TCI | PCI | VSWI | LAI | ET | SMCI | VHI |
|---|---|---|---|---|---|---|---|---|
| SPEI-1 | 0.082 | 0.362 | 0.581 | 0.065 | 0.079 | 0.035 | 0.412 | 0.114 |
| SPEI-3 | 0.131 | 0.344 | 0.542 | 0.088 | 0.117 | 0.046 | 0.396 | 0.145 |
| SPEI-6 | 0.232 | 0.238 | 0.421 | 0.149 | 0.184 | 0.059 | 0.367 | 0.189 |
| SPEI-12 | 0.261 | 0.172 | 0.311 | 0.196 | 0.227 | 0.121 | 0.302 | 0.238 |

The above analysis shows that the individual remote sensing drought indices of PCI, VCI, TCI, SMCI, VSWI, LAI, ET, and VHI have limitations in monitoring drought. Although the correlation between PCI and SPEI is high, a single precipitation factor cannot provide accurate drought information. Therefore, applying advanced deep learning methods to fuse multiple drought-causing factors is important to build a comprehensive model for monitoring drought.

### 2.9. Calibration of the Model

In the RF construction process, the number of decision trees and the maximum number of features considered for decision partitioning are important parameters that affect the predictive power of the stochastic Sen model. In addition, the RF model in this study was built using the integrated RF algorithm module in sklearn in Python. The RF model was debugged several times, and the final decision tree was chosen to be 500, the maximum tree depth was 7, and the maximum number of features was 0.8. Both XGBoost and SVM had default settings.

During the construction of the four deep learning models, each parameter of the proposed deep learning model was modified by adjusting the hidden layer, neurons, learning rate, and iteration period to obtain acceptable results. The parameter settings for the four deep learning model runs are shown in Table 5. A dropout layer was added between the model input and hidden layers to avoid overfitting and to improve the generalization ability of the models. Adam was used as the optimization algorithm for the gradient update [57]. To improve the generalization ability of the model, a non-linear Relu activation function was added between each network layer of the model. In addition, the learning rate, which is an important hyperparameter in the model construction process that determines the rate of the gradient update of the model parameters, was finally taken to be 0.001 after several experiments. The mean square error (MSE) was then used as the loss function of the model to reflect the difference between the target value and the model output value. The model metrics also used MAE to monitor the performance of the model in the test set. A number of experiments were conducted to adjust the model

run parameters to achieve optimal results. Finally, these algorithms were used to build a comprehensive drought monitoring model using the Keres framework based on Python.

**Table 5.** Four deep learning model parameter settings.

| Parameter | DFNN Value | CNN Value | LSTM Value | ConvLSTM Value |
|---|---|---|---|---|
| Layers | 6 | 9 | 6 | 11 |
| Batch size | 10 | 10 | 10 | 10 |
| Epochs | 500 | 200 | 200 | 200 |
| Learning rate | 0.001 | 0.001 | 0.001 | 0.001 |
| Pool size | — | 1 | — | 2 |
| Dropout | 0.2 | 0.2 | 0.2 | 0.2 |
| Optimization | Adam | Adam | Adam | Adam |
| Loss function | MSE | MSE | MSE | MSE |
| Activation function | Relu | Relu | Relu | Relu |
| Metrics | MAE | MAE | MAE | MAE |

## 3. Results

### 3.1. Comparison of Simulation Accuracy of Seven Models

In order to assess the monitoring capability of the model in Xinjiang, the performance metrics of the seven models on the test set for the four scales of SPEI were analyzed, and the statistical results are shown in Table 6. The results show that the ConvLSTM model simulated the largest correlation coefficient between the fitted CDI and the measured SPEI, and its RMSE and MAE values were the smallest. In addition, at the SPEI-12 scale, the ConvLSTM model output an $R^2$ of 0.874, an RMSE of 0.365, and an MAE of 0.265 between the CDI and the SPEI. The ConvLSTM also exhibited the highest monitoring accuracy at the 1-, 3-, and 6-month scales. Therefore, the ConvLSTM model had an advantage over its counterparts and met the best model criteria. In addition, CDI showed higher monitoring performance on the 12-month scale than on the 6-, 3-, or 1-month scales. This result may be due to the smoother time series in SPEI-12 compared to SPEI-6, SPEI-3, and SPEI-1, resulting in higher monitoring accuracy.

**Table 6.** Monitoring performance metrics for the seven models on the test set.

| Model | Index | SPEI-1 | SPEI-3 | SPEI-6 | SPEI-12 |
|---|---|---|---|---|---|
| RF | $R^2$ | 0.227 | 0.564 | 0.624 | 0.751 |
| | RMSE | 0.996 | 0.722 | 0.971 | 0.515 |
| | MAE | 0.809 | 0.561 | 0.522 | 0.398 |
| SVM | $R^2$ | 0.078 | 0.498 | 0.547 | 0.681 |
| | RMSE | 1.034 | 0.791 | 0.739 | 0.592 |
| | MAE | 0.824 | 0.598 | 0.569 | 0.451 |
| XGBoost | $R^2$ | 0.132 | 0.516 | 0.598 | 0.726 |
| | RMSE | 1.016 | 0.781 | 0.668 | 0.559 |
| | MAE | 0.878 | 0.572 | 0.531 | 0.422 |
| DFNN | $R^2$ | 0.322 | 0.583 | 0.632 | 0.801 |
| | RMSE | 0.868 | 0.716 | 0.633 | 0.432 |
| | MAE | 0.692 | 0.554 | 0.499 | 0.344 |
| CNN | $R^2$ | 0.371 | 0.577 | 0.693 | 0.827 |
| | RMSE | 0.848 | 0.719 | 0.568 | 0.414 |
| | MAE | 0.659 | 0.558 | 0.433 | 0.321 |
| LSTM | $R^2$ | 0.359 | 0.559 | 0.686 | 0.819 |
| | RMSE | 0.855 | 0.725 | 0.590 | 0.421 |
| | MAE | 0.671 | 0.562 | 0.449 | 0.331 |
| ConvLSTM | $R^2$ | 0.423 | 0.613 | 0.723 | 0.874 |
| | RMSE | 0.812 | 0.671 | 0.561 | 0.365 |
| | MAE | 0.623 | 0.522 | 0.424 | 0.265 |

In summary, the ConvLSTM proposed in this study proved to be a robust drought monitoring model that captured fluctuating trends in the CDI of each drought impact factor based on its previous record with the SPEI series, indicating that the ConvLSTM is a promising drought monitoring model. The CDI output from the model was subsequently used for validation against other indices.

### 3.2. Drought Consistency Analysis

According to the drought classification of the SPEI index (Table 3), the number of stations with SPEI values and ConvLSTM model monitoring values of CDI for each drought class at 55 stations from 2000 to 2020 were counted in this paper, and the results are shown in Table 7. The drought-free and light drought classes of the SPEI and CDI for all four scales were consistent with each other more than 80% of the time. The consistency between SPEI and CDI for all scales reached over 90%, except for the medium drought class of SPEI-1. For the severe drought scale, the CDI consistency rate for SPEI-6 and SPEI-12 reached over 95%, while the rates for SPEI-1 and SPEI-3 were 58.13% and 68.86%. The four scales of SPEI had the lowest consistency rate for extreme drought, with only 35.56% for SPEI-1, which may be related to the relatively low number of exceptional droughts during the 20-year period. Overall, the CDI values were consistent with the drought categorization from SPEI.

**Table 7.** Drought categorization consistency rate between CDI and SPEI at each scale.

| Consistency Rate | SPEI-1 | SPEI-3 | SPEI-6 | SPEI-12 |
| --- | --- | --- | --- | --- |
| No drought | 86.45% | 88.58% | 92.36% | 97.01% |
| Light drought | 96.73% | 82.03% | 83.86% | 97.67% |
| Moderate drought | 84.62% | 92.69% | 97.67% | 97.12% |
| Severe drought | 58.13% | 68.86% | 95.12% | 96.51% |
| Extreme drought | 35.56% | 44.81% | 76.82% | 66.46% |

### 3.3. Correlation Analysis Based on Meteorological Drought Indices

To verify the ability of the model's CDI output to monitor information from other drought indices, this study used SPI values for 55 meteorological stations in the Xinjiang Uygur Autonomous Region from 2015 to 2020. The SPI values were calculated from station precipitation, and different time scales of SPI implied different physical significances. The shorter time scales indicated changes in soil moisture, which is important for agricultural production. The longer time scale reflected long-term runoff changes, which are of practical value for reservoir management. Thus, SPI-1 and SPI-3 are good indicators of short-term changes in agricultural drought characteristics. SPI-6 is a significant marker of drought occurrence and persistence. SPI-12 is a good indicator of the effect of precipitation on changes in soil moisture and groundwater quantity [58]. In the Xinjiang region, SPI has been shown to be an effective and accurate meteorological drought index for assessing and monitoring drought [59]. In this study, a correlation analysis was conducted between CDI and SPI at similar time scales (Figure 4). The CDI obtained from the ConvLSTM model showed highly significant positive correlations with all four scales of SPI, and the correlation coefficients of the four scales of CDI for SPI-1, SPI-3, SPI-6, and SPI-12 were all higher than 0.5, passing the $p < 0.01$ significance test. This indicates that CDI can monitor short-term agricultural droughts and long-term hydrological droughts.

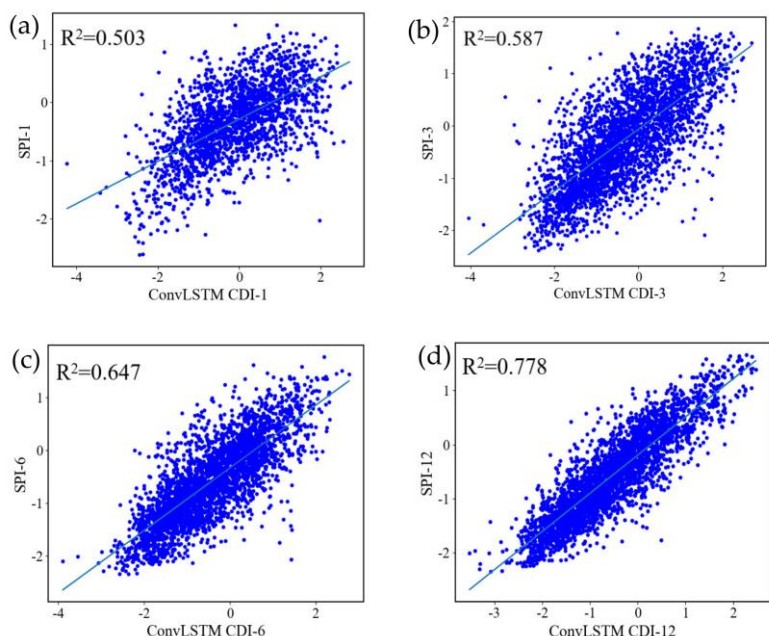

**Figure 4.** Scatter plots at four scales (**a**) SPI-1, (**b**) SPI-3, (**c**) SPI-6, (**d**) SPI-12 with the corresponding scale CDI output from this model.

### 3.4. Correlation Analysis Based on Relative Soil Moisture

To validate the applicability of the ConvLSTM model for agricultural drought monitoring, nine soil moisture sites with uniform distributions and long time series were selected to validate the reliability of the integrated drought monitoring model. The soil moisture site information is shown in Table 8. A correlation analysis was conducted using the monthly soil relative humidity at a 10 cm depth from 2000 to 2013 with the CDI-1 index output by the model. The results are shown in Figure 5. The results show that the correlation coefficients for all nine sites were above 0.45, which was significant at a 99% significance level. The monitored values correlated well with the soil relative humidity, with correlation coefficients ranging from 0.457 to 0.759. The correlation coefficient of Xinyuan station was the highest, at 0.759; the correlation coefficient of Yining station was the smallest (0.457); and the correlation coefficients of all other stations were above 0.45. Therefore, the variation in CDI from the ConvLSTM model constructed in this study can reflect variation in regional soil relative humidity. Since soil relative humidity is an important factor affecting agricultural drought, the model also has some monitoring ability for regional agricultural drought.

**Table 8.** Soil moisture site information.

| Station Code | Station Name | Latitude (°N) | Longitude (°E) | Elevation (m) |
|---|---|---|---|---|
| 51133 | Tacheng | 83.00 | 46.73 | 534.9 |
| 51379 | Jitai | 89.57 | 44.02 | 793.5 |
| 51431 | Yining | 81.33 | 43.95 | 662.5 |
| 51436 | Xinyuan | 83.30 | 43.45 | 928.2 |
| 51437 | Zhaosu | 81.13 | 43.15 | 1851 |
| 51656 | Korla | 86.13 | 41.75 | 931.5 |
| 51777 | Ruoqiang | 88.17 | 39.03 | 887.7 |
| 51811 | Shache | 77.27 | 38.43 | 1231.2 |
| 51931 | Yutian | 81.65 | 36.85 | 1422 |

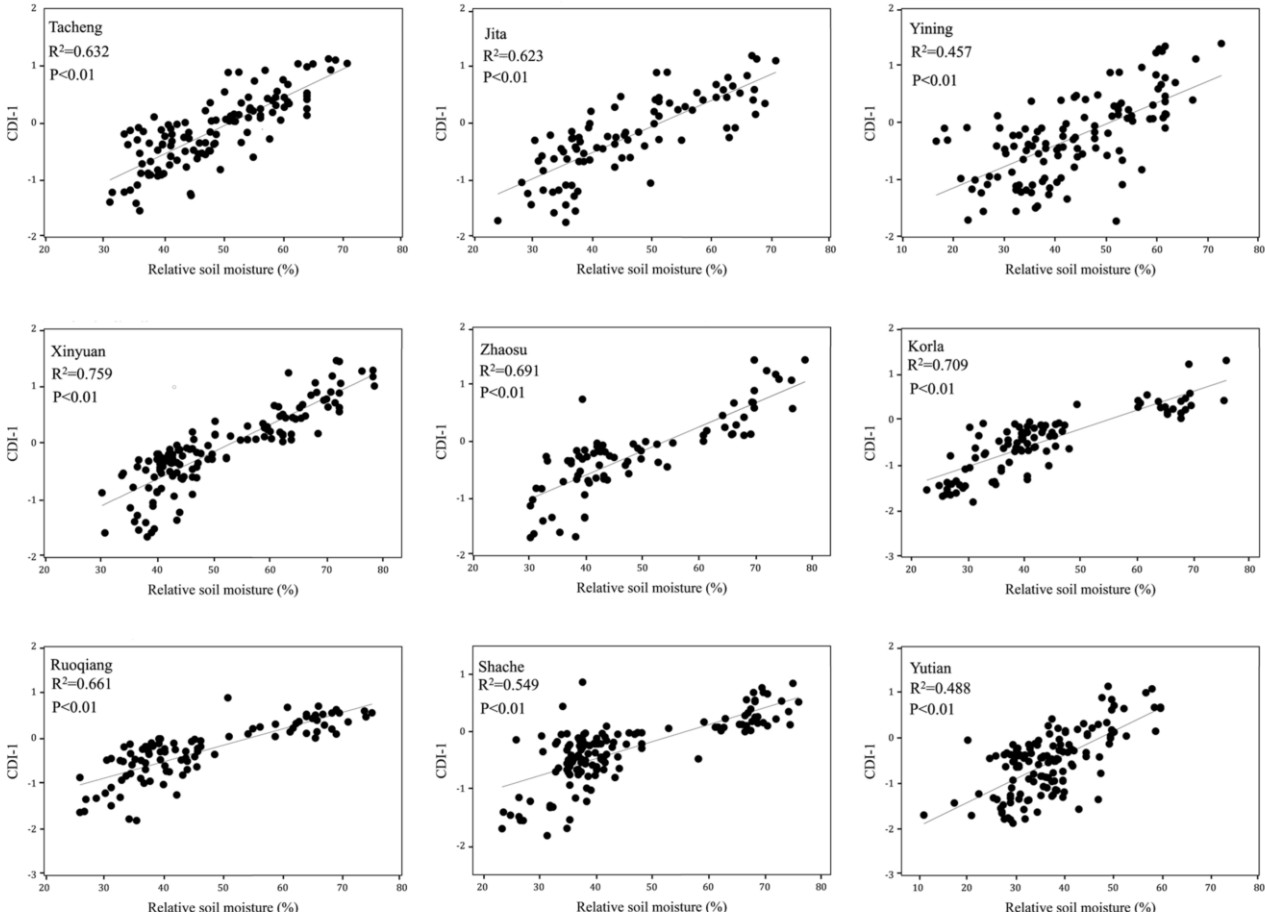

**Figure 5.** Scatter plot of CDI-1 versus the 10 cm soil relative humidity.

*3.5. Validation of the Spatial Distribution of Drought Development in a Typical Dry Year*

To further verify the spatial rationality of the CDI, we selected a typical drought year to analyze the spatial distribution characteristics of drought in Xinjiang, and we divided the CDI index into five classes, as shown in Table 9.

**Table 9.** CDI classification criteria for grading drought events.

| Drought Grade | Drought Condition | CDI |
|:---:|:---:|:---:|
| I | No drought | $0 < CDI$ |
| II | Light drought | $-0.5 < CDI \leq 0$ |
| III | Moderate drought | $-1 < CDI \leq -0.5$ |
| IV | Severe drought | $-1.5 < CDI \leq -1$ |
| V | Extreme drought | $CDI \leq -1.5$ |

According to the Xinjiang Climate Bulletin and Impact Assessment issued by the Xinjiang Uygur Autonomous Region Meteorological Bureau (http://xj.cma.gov.cn/) (accessed on 24 May 2022), the precipitation in 2014 was slightly less in most of Xinjiang, with major catastrophic weather and climate events occurring on an overall larger scale. From December 2013 to May 2014, the average precipitation in the southern Xinjiang region was 4.2 mm, nearly 80% less than normal and the lowest since meteorological records began. In addition, counties and cities such as Luntai and Ruoqiang had no precipitation for six consecutive months, with drought conditions easing after May. Regional drought events occurred in both summer and autumn, with the drought area shifting northwestward and then shifting southward in the winter. Direct economic losses were caused by various meteorological disasters, with drought disasters being the largest, accounting for approximately 38% of the total losses.

In this study, ConvLSTM was used to construct a drought monitoring model to monitor and classify drought from March to August 2014 in Xinjiang. After a comparative analysis, SPEI-6 outperformed the other scales of SPEI in characterizing severe and extreme drought. Therefore, the CDI-6 spatial distribution raster data and interpolated station SPEI-6 data were selected to evaluate the accuracy of the CDI-6 spatial distribution in monitoring drought and the reflection of drought in off-site raster data (Figure 6). The degree of drought expressed by the CDI-6 spatial distribution and SPEI observations at the actual station remained largely consistent. In March 2014, the eastern and central regions of Xinjiang experienced severe drought conditions, with 60% of the region affected by severe or extreme drought. By April, the drought had abated overall, with moderate and severe drought dominating, mainly due to the extremely low precipitation in the region during the first five months. In May, there was widespread precipitation in southern Xinjiang, which reduced the drought conditions and shifted the drought trend to the west and southwest, with drought conditions in the east and central regions showing an easing trend. In June, the drought trend shifted to the northwest and north. In July, the drought trend continued northward, with a downward trend in the severity of the drought in the south. By August, the severity of the drought in the north was reduced due to increased rainfall in the north, and the drought in the south was largely lifted. In addition, in 2014, mild drought occurred, mainly in the eastern region, while the southern region was dominated by moderate drought. Severe drought mainly occurred in the central and northwestern regions. Overall, the spatial distribution of CDI-6 from the ConvLSTM model was generally consistent with the degree of drought from the SPEI-6 observations, and the monitoring results were more detailed than the interpolated results.

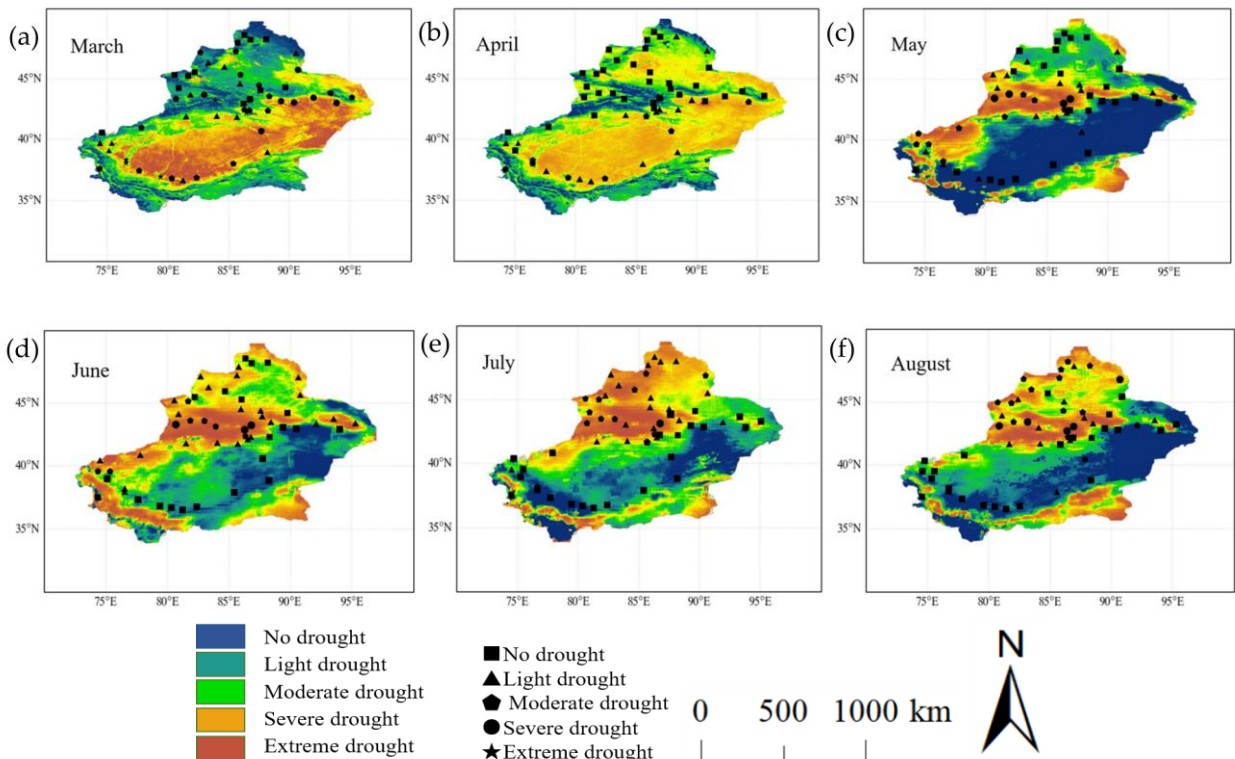

**Figure 6.** (**a**–**f**) are Spatial distribution of CDI-6 from ConvLSTM model with station drought distribution during March–August 2014.

### 3.6. Relative Importance of Different Influencing Factors on Simulation Results

Drought is influenced by a variety of factors. In order to investigate the influence of different influencing factors on the monitoring results, this study entered different influencing factors as independent model variables and used the mean reduction in accuracy as a

standard metric to obtain the relative importance of each drought factor on the simulation results. The results are shown in Table 10. On the four scales, the relative importance of PCI was 28.52, 22.93, 34.77, and 40.61%, respectively, which was the highest relative importance among the eight influencing factors, indicating that precipitation is the most important influencing factor affecting drought in the model. This was followed by TCI, VCI, and VHI, so surface temperature and vegetation factors also play important roles in the simulation of CDI in the model. The relative importance of the remaining drought-influencing factors was less than 10%, indicating that these influencing factors contribute relatively little to the monitoring results. The relative importance of ET was the lowest, probably due to the high number of missing values in the ET data from MODIS, which reduced the accuracy of the data to some extent, indicating that drought is less influenced by ET than other influences in this study area.

**Table 10.** Relative importance of factors for drought assessment.

| Impact Factors | Relative Importance (%) | | | |
|:---:|:---:|:---:|:---:|:---:|
| | CDI-1 | CDI-3 | CDI-6 | CDI-12 |
| PCI | 28.52 | 22.93 | 34.77 | 40.61 |
| TCI | 17.81 | 14.73 | 13.38 | 12.33 |
| VCI | 8.85 | 21.49 | 11.96 | 8.65 |
| VHI | 19.48 | 14.84 | 11.85 | 11.68 |
| VSWI | 6.06 | 7.05 | 7.88 | 8.58 |
| LAI | 4.41 | 6.26 | 7.81 | 6.31 |
| SMCI | 10.53 | 8.45 | 7.76 | 5.95 |
| ET | 4.34 | 4.25 | 4.59 | 5.89 |

## 4. Discussion

In this study, we proposed the ConvLSTM model for drought monitoring and achieved good monitoring results. Many previous studies have developed drought monitoring models, but most have only used relatively simple machine learning methods with relatively limited exploration of deep learning. In this study, multi-source remote sensing and station data were used as input data to construct a comprehensive drought index using deep learning algorithms, and the monitoring results were more detailed than traditional interpolation results. The good local features make this model superior to traditional meteorological monitoring methods. However, despite the novelty of the proposed model, it still has some limitations. Firstly, in terms of model structure, ConvLSTM reflects the time series relationship of drought well and can effectively capture the complex relationships between weather indices and remote sensing indices. However, it is more complicated than traditional machine learning methods in terms of memory and running time when optimizing parameters and requires iterative debugging to obtain better results. Secondly, deep learning methods are often referred to as black box models and give results that are difficult to interpret in terms of causal relationships between variables and specific importance.

There were also some limitations in terms of data. Firstly, when selecting remote sensing data to calculate the drought index, different remote sensing data had different spatial resolutions, which may have caused data bias and increased uncertainty, thus further affecting the model monitoring performance. Secondly, it was difficult to obtain soil moisture data with higher accuracy for long time series. The station soil moisture data obtained in this study were from 2000 to 2013, and there were data deficiencies, making the correlation of CDI to soil relative humidity validation results slightly lower than those of the meteorological drought index. Thirdly, the time scale of the constructed integrated drought monitoring model was monthly, which did not reflect the influence of specific time on drought development. The ability to monitor drought in real-time is yet to be studied. Fourthly, we calculated the proportions of land cover types in the study area, and the results showed that the land cover types in the study area were relatively simple. As of 2020, deserts accounted for 67.57% of the total area, forests and grasslands accounted

for 25.88% of the total area, and the remaining land types accounted for 6.55% of the total area, indicating that the Xinjiang region is dominated by deserts and that the impact of vegetation on drought is relatively small. Moreover, we used station data for validation. The distance between stations varied greatly, and the climate type varied greatly from station to station in the same month, which is an important reason for the low accuracy of short-term drought monitoring.

Finally, this study only considered the influencing factors of precipitation, vegetation, surface temperature, and soil moisture in the selection of influencing factors to construct the drought monitoring model. Due to the complexity of drought-influencing factors, topography, vegetation type, surface albedo, and soil water availability capacity also influence drought. These factors will be considered in future research, and deep learning models that are more suitable for drought monitoring will be explored in order to monitor drought more accurately.

## 5. Conclusions

Drought is a complex natural phenomenon. This study proposed a ConvLSTM deep learning model, combined with multi-source remote sensing data, to construct a comprehensive drought monitoring model to simulate CDI to monitor drought information in Xinjiang. The results were compared with three traditional machine learning models and three different types of deep learning models, and the monitoring performance was evaluated by three evaluation metrics. The following is a summary of the study results.

The integrated drought monitoring model effectively monitored drought information in the Xinjiang region. The generalization capability of the model was improved by establishing cross-validation between the training and test sets during the model training process. The CDI obtained from the output of the ConvLSTM model was highly consistent with the measured SPEI index, with the consistency rate of each drought category exceeding 80%, except for extreme and exceptional drought. The model monitoring values were more consistent with the station-derived SPEI values.

In the correlation analysis with meteorological drought, the correlation coefficients between the model's CDI output and the 2015 station SPI data for all four scales were above 0.5 ($p < 0.01$), reaching a significant correlation level. In the correlation analysis with agricultural drought, the correlation coefficients between the model's CDI-1 output and the soil relative humidity at a 10 cm depth at agrometeorological stations were all greater than 0.45 ($p < 0.01$). This indicates that the model has good applicability in integrated drought monitoring.

In this study, based on the model to monitor the typical drought months in the Xinjiang region from March to August 2014, the CDI-6 spatial distribution plotted according to the model was generally consistent with the drought conditions reflected by the station observations. The drought conditions shown by the raster data of the CDI-6 spatial distribution outside the stations were consistent with the actual recorded drought conditions, which can reflect drought development and spatial evolution.

**Author Contributions:** Conceptualization, D.X. and Y.Z.; methodology, D.X.; validation, D.X., Y.Z. and W.T.; formal analysis, D.X.; investigation, D.X.; resources, Y.Z.; data curation, D.X.; writing—original draft preparation, D.X.; writing—review and editing, Y.Z., K.T.C.L.K.S. and W.T.; visualization, D.X.; supervision, W.T., H.Z., S.G., H.L., G.M. and J.H.; project administration, Y.Z.; funding acquisition, Y.Z. All authors have read and agreed to the published version of the manuscript.

**Funding:** This research was funded by the National Key Research and Development Program of China (grant No. 2021-YFE0116900); the National Natural Science Foundation of China (grant No. 42175157); the Fengyun Application Pioneering Project (FY-APP) (grant No. FY-APP-2022.0604); and in part by the Postgraduate Research and Practice Innovation Program of Jiangsu Province (grants KYCX21_0997 and SJCX22_0354).

**Data Availability Statement:** The data presented in this study is available by following the links in the article.

**Acknowledgments:** Thanks to the National Meteorological Administration of China for providing the meteorological data for this study.

**Conflicts of Interest:** The authors declare no conflict of interest.

## Appendix A

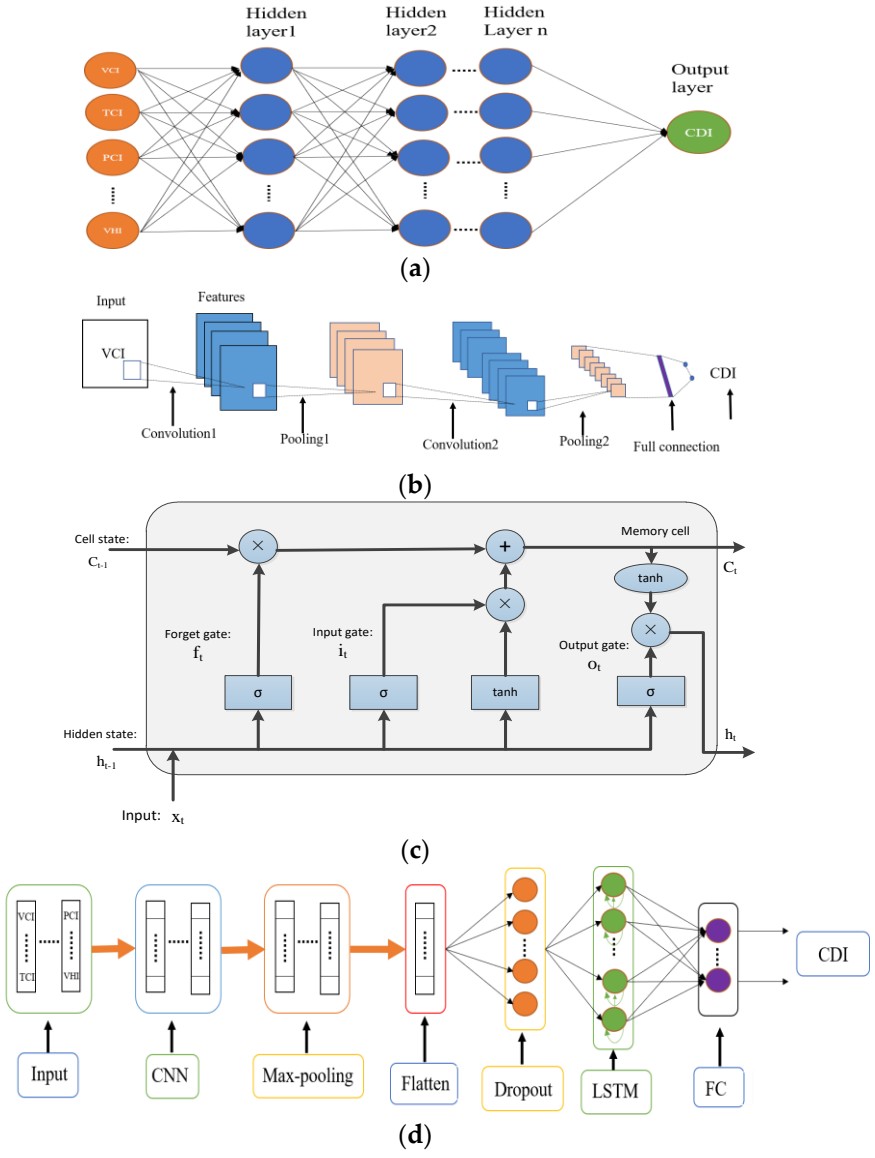

**Figure A1.** (**a**) Architecture of the DFNN model used in this study; (**b**) CNN model architecture; (**c**) architecture of the LSTM model; and (**d**) overall structure of the ConvLSTM network.

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
