# Peer review of "Construction of an Integrated Drought Monitoring Model Based on Deep Learning Algorithms"

_remotesensing, doi:10.3390/rs15030667_

Round 1

Reviewer 1 Report (Previous Reviewer 2)

the manuscript needs english polish

Author Response

We have made changes to the language

Reviewer 2 Report (Previous Reviewer 4)

All the questions  have been modified

Author Response

Thank you for your advice

Reviewer 3 Report (Previous Reviewer 3)

See attachment

Round 2

Reviewer 1 Report (Previous Reviewer 2)

the manuscript was well revised and it can be accepted for publicaiton.

Author Response

Thank you again for your comments

Reviewer 3 Report (Previous Reviewer 3)

No additional comments. My concerns were addressed.

Author Response

Thank you again for your comments

This manuscript is a resubmission of an earlier submission. The following is a list of the peer review reports and author responses from that submission.

Round 1

Reviewer 1 Report

Dear Authors,

First of all thank you for your paper. Before accepting your paper to publish, I believe there are some points to be clarified (or improved). I believe, probably due to the nature of the topic, the paper is somehow confusing. As an example, in Table 2, for the PCI, you stated that, Pi is the precipitation in a month and Pmax and Pmin are minimum and maximum precipitation in a month. I can only assume that, you used daily precipitation values, and min and max values are the minimum and maximum daily values within a month. However, as you mentioned in the previous section, you have monthly precipitation data (if I did not misunderstand). This situation is more confusing for VCI. Please explain the variables more clearly.

Please write the units of each drought variable (please write unitless if they are).

In Table 3, you represented the drought intervals for SPEI model outputs. Moreover, in in table 5, you gave the results of the models. As far as I could understand, the RMSE values are to high for a range between 0 and 2 (or -2). If the results are related with other models please state them as well.

Finally, to sum up, please clarify your sentences and use more explicit ways to explain you study.

Reviewer 2 Report

The entitled "Deep learning comprehensive drought monitoring model construction based on multi-source remote sensing data" tried to generate the CDI for monitoring drought in xinjiang using deep learning and machine learning methods. The topic is interesting, but there lacks significant meaning for doing so. I do not recommend the publication of this manuscript.

line 93 and 94. The refered reference was not appropriate. It can not support the effectiveness of deep learning for monitoring droughts. Check the whole manuscript, and avoid all similar use.

line 100 and 101. There have been many studies using deep learning and multi-sources of data and strong feature extraction. You hav just mentuoned Shen in line 98 using deep learning with multi-source of remote sensing data. The current expression is not appropriate. Redo the review of this field.

line 118 , why this period?

line 123. Revise the format of KM2. Keep the longitude and latitude to the minute precision.

line 131. The authors should mention the xinjiang is lacks of water and it is one of the driest regions in the world in Section introduction.

table 1. the spatial resolution of each data should be given.

figure 2. mark clearly which part are the cnn and lstm in?

why only the random forest was selected as the traditional machine learning approaches? The commonly applied SVM, XGBOOST, ANN(BP) can also be tested if the authors tried to prove the deep learning methods were better. Besides, the results in table 5 already showed the RF had achieved relatively high accuracy. Considering the calculation efficients, I think RF was better as it can save time, and it do not the CNN-based workflow.

line 319 to 328. The definition of r2 rmse, and MAE can be deleted as these were commonsense.

The CNN-based deep learning approaches can extract massive useful information through the training dataset, and these data was commonly image-based ones. The authors calculated the VIs and applied these as independent variables. The useful informtion from the whole image was likely ignored, and thus, I do think the CNN-LSTM could achieve a better regression as the real mechanism is the same as machine learning based methods.

figure 5. there is no significant difference for SPI-3 and SPI-6 obtained from different deep learning and machine learning approaches.

The analysis of drought is a long time event, and the authors only assessed the drought in xinjiang with limited months of data, the number of samples were real small actually. The results were not reliable. 

The authors directly applied different VIs and climatic variables for generating comprehensive drought index (CDI), and found the CDI was closely correlated with SPEI in differnt time scales. As mentioned in the article, both the CDI and SPEI were obtained based on climatic data, and thus, there is no doubt they have highly correlated. I suggest before generating the CDI, the authors should calculate the relationships between the VIs, climatic variables and SPEI. For the current form, there lacked a scientific meaning for doing this.

Reviewer 3 Report

See attachment

Reviewer 4 Report

Comments: This paper constructed an integrated drought monitoring model (comprehensive drought index) based on the CNN-LSTM in the Xinjiang. Although this could be an interesting topic for remote sensing, the manuscript shows several shortcomings that can't be ignored:

1. Introduction. Except the drought monitoring based on the meteorological or remote sensing data, other technical method (VIC, SWAT and other hydrologic model) should be added.

2. Line 55-56 was the improper description. In fact, the SPEI data with 0.25°× 0.25°can be download from http://digital.csic.es/handle/10261/153475. This data was calculated by CRU data.

3.  Line 74. The shortcoming of construct drought monitoring models by linear model should be add.

4. Line 96 was the improper description. In fact, some studies had been using deep learning algorithms for drought monitoring.

Construction of a drought monitoring model using deep learning based on multi-source remote sensing data

5. Line 168. Resample generally means from high resolution to low resolution. The authors resampled 0.25°to 500m is improper in this manuscript.

6. Line 173. The data are for the period 2000-2014, why not choose the period 2000-2020 or 2021?

7. SPEI-1 not choose in this manuscript, why? Many drought events will not last three months.

8. Why not choose the soil moisture to verify the CDI?

9. The discussion was not deep enough

10. Conclusions are too long and authors must be simplified. 

Round 2

Reviewer 1 Report

Dear Authors,

Thank you for the kind responses to the questions.

Reviewer 2 Report

I could see the manuscript was improved. But it is still far from being accepted for publications. Firstly, the commonly applied machine learning can be added for comparison, which was more convincing. The authors did not get the point of deep learning as there is no need to calculate the VIs. The manuscript still need to be improved, and it cannot be accepted for publication for the current status.
